# Assessment of Microvascular Disease in Heart and Brain by MRI: Application in Heart Failure with Preserved Ejection Fraction and Cerebral Small Vessel Disease

**DOI:** 10.3390/medicina59091596

**Published:** 2023-09-04

**Authors:** Jonathan Bennett, Maud van Dinther, Paulien Voorter, Walter Backes, Josephine Barnes, Frederick Barkhof, Gabriella Captur, Alun D. Hughes, Carole Sudre, Thomas A. Treibel

**Affiliations:** 1Institute of Cardiovascular Science, University College London, London WC1E 6BT, UK; 2Department of Cardiology, Barts Heart Centre, London EC1A 7BE, UK; 3Department of Neurology, Maastricht University Medical Center, 6229 HX Maastricht, The Netherlands; 4School for Cardiovascular Diseases, Faculty of Health, Medicine and Life Sciences, Maastricht University, 6211 LX Maastricht, The Netherlands; 5Department of Radiology and Nuclear Medicine, Maastricht University Medical Center, 6229 HX Maastricht, The Netherlands; 6School for Mental Health & Neuroscience, Faculty of Health, Medicine and Life Sciences, Maastricht University, 6200 MD Maastricht, The Netherlands; 7Dementia Research Centre, UCL Queens Square Institute of Neurology, University College London, London WC1E 6BT, UK; 8Department of Radiology & Nuclear Medicine, Amsterdam UMC, Vrije University, P.O. Box 7057, 1007 MB Amsterdam, The Netherlands; 9Queen Square Institute of Neurology, University College London, London WC1E 6BT, UK; 10Centre for Medical Image Computing, University College London, London WC1E 6BT, UK; 11Medical Research Council Unit for Lifelong Health and Ageing, Department of Population Science and Experimental Medicine, University College London, London WC1E 6BT, UK; 12Centre for Inherited Heart Muscle Conditions, Cardiology Department, The Royal Free Hospital, London NW3 2QG, UK; 13School of Biomedical Engineering and Imaging Sciences, King’s College London, London WC2R 2LS, UK

**Keywords:** heart failure with preserved ejection fraction, cerebral small vessel disease, microvascular, cardiovascular magnetic resonance, magnetic resonance imaging, neuroimaging

## Abstract

The objective of this review is to investigate the commonalities of microvascular (small vessel) disease in heart failure with preserved ejection fraction (HFpEF) and cerebral small vessel disease (CSVD). Furthermore, the review aims to evaluate the current magnetic resonance imaging (MRI) diagnostic techniques for both conditions. By comparing the two conditions, this review seeks to identify potential opportunities to improve the understanding of both HFpEF and CSVD.

## 1. Introduction

There has been significant advancement in the diagnosis and management of heart and brain diseases over the past decades; however, much of this progress has been on macrovascular diseases of the large arteries and veins [1,2]. Microvascular disease (small vessel disease), affecting the small arteries, arterioles, capillaries, and venules of the heart and brain, presents a significant burden on patients and healthcare systems with costs totalling billions each year [3,4], but remains incompletely understood. Damage to the microvasculature can result in acute and chronic hypoperfusion that, depending on the end organ, can differentially manifest in a variety of symptoms and disorders.

Heart failure with preserved ejection fraction (HFpEF) and cerebral small vessel disease (CSVD) are complex conditions that present significant ongoing challenges in both diagnosis and treatment. There are currently limited treatments for either condition, and the lack of effective interventions may be due in part to difficulties understanding the underlying pathophysiology. Additionally, designing clinical trials for HFpEF and CSVD has proven challenging [5,6] due to the heterogeneity of the patient populations and the lack of reliable biomarkers. These factors highlight the need for continued research into the underlying mechanisms of microvascular disease in HFpEF and CSVD, as well as the development of reliable and validated diagnostic and therapeutic strategies.

The objective of this review is to investigate the commonalities of microvascular disease in HFpEF and CSVD. Furthermore, the review aims to evaluate the current magnetic resonance imaging (MRI) diagnostic techniques for both conditions. By comparing the two conditions, this review seeks to gain a greater understanding of both and identify potential opportunities to refine diagnostic approaches.

## 2. Background

### 2.1. HFpEF: Epidemiology and Clinical Presentation

HFpEF has been described as the, “single largest unmet need in cardiovascular medicine,” and comprises 50% of HF cases and admissions [7,8]. HFpEF is currently defined as the signs and symptoms of heart failure (HF), left ventricular ejection fraction (LVEF) > 50%, and objective evidence of cardiac structural and/or functional abnormalities consistent with diastolic dysfunction, raised filling pressures, or elevated natriuretic peptides [8]. Patients present with cardinal features of HF including fatigue, dyspnoea, and ankle swelling that is often accompanied by clinical signs of elevated jugular venous pressure, and pulmonary and peripheral oedema.

The echocardiographic cutoff LVEF > 50% is somewhat arbitrary (and varies depending on the imaging modality used) and it is recognised that systolic dysfunction is present in HFpEF; however, LVEF is relatively insensitive to detect mild impairment. Furthermore, recent data [9] suggest that there are multiple subcategories within the HFpEF population and other cardiovascular diseases (cardiac amyloidosis, hypertrophic cardiomyopathy, and constrictive pericarditis) can present with HFpEF [10] but have an alternative underlying pathophysiology with a specific therapeutic pathway. Serial myocardial biopsy study in HFpEF [11] demonstrated 14% prevalence of cardiac amyloidosis, with further cardiovascular magnetic resonance (CMR) study identifying alternative pathology in more than a quarter of individuals presenting with HFpEF [10]. These studies highlight the need for improved phenotyping and may in part explain the poor outcome of previous HFpEF trials, as the standard echocardiography used as the primary enrolment modality struggles to rigorously exclude cases of HFpEF with an alternative underlying mechanism that would not respond to the trial intervention.

When compared to HF with reduced ejection fraction, HFpEF patients are typically older, female, with a higher burden of diabetes, chronic kidney disease, obesity, and less coronary artery disease. Prognosis is favourable compared to HF with reduced ejection fraction [12], but there remains an excess mortality and significant reductions of quality of life.

### 2.2. HFpEF: Pathophysiology

The underling pathophysiology of HFpEF is incompletely understood with multiple models developed to explain the high diastolic left ventricular stiffness that is a cornerstone finding. Paulus et al. [13] propose a systemic pro-inflammatory model, suggesting that comorbidities, such as diabetes, obesity, and hypertension, cause coronary microvascular inflammation and promote the expansion of the extracellular matrix with excess fibrotic collagen and cardiomyocyte dysfunction (hypertrophy, impaired relaxation, and increased stiffness). This model is supported by human biopsy study [14] in HFpEF patients, which shows elevated levels of pro-inflammatory, fibrotic factors, and myofibroblasts compared to controls. Also, plasma markers of inflammation (tumour necrosis factor α and interleukin-6) are associated with incident risk of HFpEF [15].

Complex imbalances in nitric oxide pathways associated with both pro-inflammatory stress and mechanical stress (from hypertension and arterial stiffness) [16] are believed to play a key role in HFpEF. Schiattarella et al. [16] recently demonstrated a model in which HFpEF can be induced in mice with a high-fat diet and nitric oxide synthase inhibition. This may paradoxically increase the activity of inducible nitric oxide synthase, which produces large quantities of nitric oxide [17] that may drive cardiomyocyte dysfunction. Specific inhibition of inducible nitric oxide synthase led to a less severe HFpEF phenotype in their mouse model. Histologically, they describe cardiomyocyte hypertrophy, extracellular matrix fibrosis, and disruption of myocardial capillaries with thickening of the capillary basal lamina, oedematous endothelial cells, and capillary lumen narrowing and irregularities with defective attempts at angiogenesis also leading to reduced density (capillary rarefaction) and quality of vessels and, therefore, decreased myocardial oxygen uptake [16,18,19].

Shiattarella et al. [16] demonstrated increased aortic stiffness and impaired coronary artery endothelial function in their mouse model; this has long been hypothesized as a contributing factor to the development of HFpEF. Elevated arterial stiffness is a key feature in HFpEF and is elevated higher than age-matched controls and hypertensives [20]. This elevated arterial stiffness, especially upon exercise [21], increases arterial pulse pressure and afterload. It is hypothesised that the stiff arteries cannot adequately absorb the pulse energy, and therefore, the excess energy is transferred through to the end organ microvascular bed and tissue [22]. Additionally, the increased arterial stiffness leads to early arrival of reflected waves to the heart increasing mid-to-late systolic load on the LV [23]. As the reflected wave arrives early in systole, it cannot contribute to diastolic pressure; therefore, coronary perfusion is impaired that may promote further ischaemia and tissue damage [24].

### 2.3. HFpEF: Non-MRI Assessment of Microvascular Disease

Several modalities, not limited to CMR, are available to assess microvascular disease in the heart. Myocardial biopsy is not routinely undertaken in those presenting with HFpEF, and to date, no modality can directly visualize the coronary microvascular system in vivo. Therefore, current assessment is limited to indirect functional assessment of microvascular function or examination of the downstream structural effects of microvascular disease on the heart.

Echocardiography is the standard first-line investigation that assesses structural and functional alterations of HFpEF including left ventricular structure (LVH and remodelling) and function (diastolic relaxation and strain), left atrial dilatation, and elevated pulmonary pressures that occur in response to increased filling pressures. Echocardiography has been used to estimate myocardial blood flow (MBF) through Doppler assessment of the left anterior descending artery on the parasternal short-axis view during hyperaemia and rest, allowing the calculation of myocardial perfusion reserve (MPR) as the ratio between stress and rest. In the PROMIS-HFpEF study [25], 75% of patients had reduced MPR (<2.5) assessed via this method, and this was associated with increased natriuretic peptides’ reduced longitudinal strain. Echocardiography is advantageous for its widespread availability and inclusion in guidelines, harmonization between vendors (limited in strain imaging), and high temporal resolution for diastology. However, it is limited by its imaging windows that are often impaired in the presence of comorbidities (obesity, chronic lung disease), MPR assessment not being standard workflow, and limited tissue characterization. Nearly 20% of the PROMIS-HFpEF cohort could not have MPR assessed due to these issues.

Stress positron emission tomography (PET) is a well-established method of estimating myocardial blood flow MBF and MPR and has shown similar results to echocardiography with MPR being reduced when compared to controls and hypertensive LVH patients [26]. PET is limited by low spatial resolution, no tissue characterization, and ionizing radiation, making it not suitable for longitudinal studies.

Invasive coronary angiography is advantageous in excluding significant coronary artery disease and consistently demonstrates impaired MPR (often termed coronary flow reserve) [27] and elevated index of microcirculatory resistance (which is not available on other modalities). It has elevated patient risk due to its invasive nature, and this potentially introduces bias as patients are often opportunistically recruited when referred for angiography for a clinical indication.

### 2.4. CSVD: Epidemiology and Clinical Presentation

Imaging markers of CSVD (Table 1) are presumed to be the result of damage to the small blood vessels in the brain. They are intimately associated with the development of vascular cognitive impairment and stroke, both ischaemic and haemorrhagic, but can also be seen in cognitively normal elderly individuals [4,28]. CSVD is associated with recurrent strokes, large-artery atherosclerosis [29], and dementia, where Alzheimer’s is the leading cause of dementia [4].

CSVD may be asymptomatic and only be identified incidentally on neuroimaging. CSVD has however been associated with various deleterious brain functions including mood disturbance, incontinence, motor and gait disturbance, cognitive impairment, and impaired executive function. Location of the lesion is thought to play an important role in the associated impacted function, with lesions in the left frontotemporal, right parietal, and left thalamus being at greatest risk of subsequent cognitive impairment [34]. CSVD is associated with increasing age and the presence of traditional cardiovascular risk factors [35] including hypertension, obesity, cigarette smoking, diabetes, obstructive sleep apnoea, and chronic kidney disease.

CSVD is heterogenous and often coexists with Alzheimer’s disease or other forms of dementia (mixed dementia), which can make diagnosis and trial design challenging [36,37].

### 2.5. CSVD: Pathophysiology

Despite advances in neuroimaging and biomarkers, the pathogenesis of CSVD is not well understood, and much of the knowledge has been acquired through animal models and postmortem examination [38]. Pathological findings of CSVD include loss of smooth muscle cells, lipohyalinosis, thickening of the vessel wall, and narrowing of the lumen [39]. Multiple processes are proposed to be involved in the pathophysiology of CSVD, including blood–brain barrier (BBB) dysfunction, changes in cerebral blood flow and perfusion, microvascular rarefaction, inflammatory and immunological processes, endothelial and pericyte dysfunction, and impaired clearance pathways. Yet, the exact pathways remain unclear, but all these processes relate to dysfunction of parts of the neurovascular unit. The neurovascular unit [40] is a functional–anatomical entity that is composed of microvascular endothelium, basal membrane, astrocytes, pericytes, microglia, neurons, and the extracellular matrix around the vessels, and it regulates the close relationship between metabolic demand and neuronal activity in the brain.

Disruption of the BBB [41] leads to an increased permeability and altered transport of molecules between blood and brain and vice versa. Consequently, it can result in dysfunction of the neurovascular unit, aberrant angiogenesis, brain hypoperfusion, and inflammatory responses, eventually leading to progressive synaptic and neuronal dysfunction. A reduced cerebral blood flow [42] could cause hypoxia and consequently lead to neuronal damage and reduction of microstructural integrity. Capillary architecture is disrupted in CSVD with increased tortuosity of microvessels being vulnerable to blockage by microemboli and decreased shear stress reducing nitric oxide synthase production [38]. Capillary rarefaction is hypothesised to occur due to active capillary regression [43], impaired angiogenesis [44], and endothelial dysfunction [38].

Inflammatory responses can lead to enhanced BBB disruption and direct damage of brain tissue [45]. Endothelial dysfunction can induce brain damage via different mechanisms, including loss of autoregulation and neurovascular coupling, BBB disruption, and capillary rarefaction. The reduction of release of nitric oxide is an established marker for endothelial dysfunction. The disruption of endothelial nitric oxide synthase, which is responsible for much of nitric oxide production, and oxidative stress (i.e., imbalance between formation of free oxygen species and defensive antioxidants) are believed to have a significant impact on the development of CSVD [46]. Animal models of CSVD have shown that mice deficient in nitric oxide exhibit cerebral hypoperfusion, increased oxidative stress, microinfarction, microbleeds, and BBB leakage [47]. Pericytes, which are responsible for controlling capillary tone and regulating blood flow, are reduced in models of CSVD. This loss of pericytes can result in uneven blood flow, BBB leakage, and pericytes transition to myofibroblasts responsible for scar formation [38]. Dysfunction of the brain waste clearance systems might lead to accumulation of brain waste and subsequently cause tissue damage.

As in HFpEF, increased arterial stiffness and elevated pulse pressure may play a role in transferring excess energy to the cerebral microvasculature and tissue. Increased arterial stiffness and elevated pulse pressure have been associated with an increased burden of WMH [48,49], cerebral microbleeds [22], and lacunar stroke [49].

### 2.6. CSVD: Non-MRI Assessment of Microvascular Disease

The imaging characterisation of CSVD is mainly based on morphological findings that can be made visible by widely available standardized multicontrast MRI protocols, largely avoiding the administration of gadolinium-based contrast agents. Computed tomography (CT) lacks the enhanced soft-tissue characterization of MRI but can reliably identify lacunar infarcts, cerebral atrophy, and the more advanced white matter lesions [28] and is the first-line investigation in acute stroke [35]. CT can assess CBF and in CSVD patients, but MRI is the preferred method [50] as it provides an enhanced soft-tissue assessment and estimation of CBF in a single comprehensive study without ionizing radiation and possible administration of iodine contrast medium. PET does not have the ability to identify the morphological features of CSVD; it does, however, have a broad evidence base for estimating CBF [51] and other related measures and can be useful in distinguishing CSVD from Alzheimer’s and other neurodegenerative pathology [52].

### 2.7. Similarities and Differences of HFpEF and CSVD

On a broad scale, the heart and the brain share some similarities in both relationship to their histology and vascular supply. Both are terminally differentiated organs with limited ability to replicate or regenerate in the event of injury; furthermore, they require disproportionate amounts of energy and oxygen relative to weight [38,53]. The brain, despite being 2% of bodyweight, accounts for 20% of an individual’s energy at rest, while, per gram, the heart requires the most oxygen of any organ in the body. Despite these high energy demands, both organs have limited energy stores, making them both vulnerable to acute and chronic reductions in energy supply and oxygen. The vascular supply of both is comparable to some extent with superficial major arteries that give off deep penetrating arterioles and capillaries for tissue perfusion [54]. Perfusion, however, is fundamentally different between the two organs. The contraction of the heart causes a cycle of diastolic perfusion with systolic halting or reversal of flow as the intramyocardial pressures compress the coronary vasculature, which does not occur in the brain. Another difference is the more internal blood supply of the brain, through the perforating branches of the middle cerebral arteries, which is prone to obstructions and characteristic for the deep locations of the morphologic features of CSVD.

Both HFpEF and CSVD are prevalent in the elderly population and are often linked to shared pro-inflammatory cardiovascular risk factors such as hypertension, diabetes, and obesity. A comparison in the pathophysiology uncovers some overlap, suggesting that imbalances in nitric oxide pathways may be a key mechanism in the development of these disorders. Furthermore, a common characteristic in both HFpEF and CSVD is the disturbance in the structure, function, and density of capillaries.

A key feature of HFpEF is its female preponderance [5], and in CSVD there is a tendency to faster progression in females and potentially increased WMH burden in relation to menopause and hot flushes [55,56]. Sex differences in CSVD can be challenging to study, which may be a reflection on women being less likely to be recruited to trials, and differing clinical presentation in cognitive impairment that may be overlooked [57].

### 2.8. MRI Techniques for Assessing Microvascular Disease

MRI is widely recognized as the gold-standard technique for assessing CSVD as it provides high-resolution imaging of the brain that reveals the structure changes and abnormalities associated with CSVD including recent small subcortical infarcts (formerly lacunar infarct), white matter hyperintensities (WMH), lacunes, enlarged perivascular spaces, cerebral microbleeds, and cortical cerebral microinfarction. The presumed pathophysiology and imaging features of these are summarized in Figure 1. More recently, CMR has emerged as a valuable tool for evaluating the structural, functional, and microvascular changes in the heart and has been applied in the assessment of HFpEF. The use of advanced CMR techniques including myocardial perfusion, T1 mapping, and late gadolinium enhancement (LGE) can help to assess microvascular disease and provide valuable insight into the pathophysiology of HFpEF and help guide in clinical decision-making. As previously discussed, it is superior to other modalities at excluding myocardial diseases masquerading as HFpEF [10].

### 2.9. CMR Assessment of HFpEF

#### 2.9.1. Cardiac Structure

CMR has established itself as the gold-standard technique for assessing cardiac structure due to its high spatial resolution and freedom from geometric assumptions, which makes it more sensitive to detecting the adverse remodeling seen in HFpEF and provides superior reproducibility compared to echocardiography [58]. The higher precision and reproducibility of CMR, compared to other modalities, makes it the preferred choice for trial design as it allows small sample sizes, potentially offsetting its higher cost. Sequences can be prescribed in unlimited planes for qualitative and quantitative assessment of cardiac chamber morphology and have superior contrast between the blood pool and myocardium when compared to echocardiography.

CMR is a precise and accurate method for assessing LVH, which is a characteristic of HFpEF, being observed in approximately 50–70% of patients. It is also associated with coronary microcirculatory structural and functional abnormalities [59], and an increased risk of major adverse cardiovascular events (MACE) [60]. Furthermore, advanced artificial intelligence methods of assessing LV structure not only outperform echocardiography but human analysis of CMR and can provide insight into patterns of LV remodeling present in HFpEF [61]. In a considerable proportion [62] of patients with HFpEF, the left atrium (LA) dilates, which is linked to MACE [58]. The LA dimensions may be precisely evaluated by CMR compared to echocardiography due to the absence of geometric constraints and better spatial resolution; however, there is limited consensus on the preferred method with the Simpson’s biplane reference standard not typically being part of the standard CMR protocol [63]. As with the LV, the right ventricle is more accurately assessed by CMR, and its function is frequently impaired in HFpEF.

#### 2.9.2. Cardiac Function

Although LVEF is preserved (>50%) in HFpEF, this does not exclude significant systolic dysfunction that can be identified through CMR-LV strain analysis. In a cohort of 131 patients with HFpEF, reduced global longitudinal strain was associated with increased MACE on multivariate analysis, while LVEF had no influence [64]. Impaired strain is associated with CMD [65,66] and diffuse myocardial fibrosis [67], two hallmark features of HFpEF making it a promising technique. There are various methods of CMR strain acquisition, the most common of which is feature-tracking that can be undertaken by postprocessed standard cine images collected in the majority of CMR protocols [58]. The small evidence base, differences in vendor postprocessing software, absence of robust normative values, and lack of standardized approach make it unsuitable for widespread clinical rollout [68].

Accurate assessment of diastolic function is crucial for the diagnosis of HFpEF, with increasing evidence that it is associated with microvascular disease [69,70,71]. Transmitral flow velocities generating E (passive diastolic flow) and A waves (active diastolic flow) can be determined using phase-contrast CMR and correlate fairly with reference standard echocardiography [72]. However phase-contrast CMR imaging has multiple limitations including lower temporal resolution, thus systematic underestimation of values, averaging of flows over multiple breath holds, phase-offset errors, appropriate selection of velocity encoding, and labour-intensive postprocessing [73]. CMR can assess the rapid myocardial velocities (e’) analogous with echocardiography Tissue Doppler Imaging with similar levels of agreement and have similar limitations as phase-contrast CMR, limited by temporal resolution and evidence base [74,75]. 4D-CMR shows promise as an alternative method of CMR diastolic assessment [76]; however, it requires long acquisition, specialist postprocessing, and therefore is not in routine clinical use and limited to research centres.

### 2.10. Myocardial Perfusion CMR

While many of the parameters assessed by CMR have been associated with microvascular disease, it is important to acknowledge that they do not provide direct measurement of the coronary microvasculature. However, myocardial perfusion CMR is rapidly gaining recognition as a reliable technique for diagnosing, quantifying, and monitoring CMD in patients with HFpEF.

The principle underlying myocardial perfusion CMR in HFpEF is that, when there is no significant coronary stenosis, valvular disease, or primary heart muscle disease, inducing hyperaemia through stress or exercise will increase myocardial blood flow (MBF) and that this response is reflective of the function of the coronary microcirculation. The most commonly used stress is adenosine or the A2A adenosine receptor agonist, Regadenason; however, protocols for Dobutamine and exercise exist [77]. During myocardial perfusion CMR, a series of images is obtained over at least three myocardial short-axis slices following injection of a gadolinium-based contrast agent (GBCA). Images are ECG-gated and acquired every 1–2 heartbeats (depending on heart rate) using a T1-weighted dynamic pulse sequence, with total acquisition taking ~40–60 s. As part of a standard protocol, the sequence is repeated twice—once at peak stress and again at rest.

Myocardial perfusion imaging has evolved from a qualitative visual assessment in which reduced MBF was appreciated as a hypointense, to advanced validated quantitative perfusion sequences that automatically segment the myocardium and quantify MBF (in mL/min/g) on a pixel-by-pixel basis [78,79]. With quantification, not only are absolute stress and rest MBF values obtained but the myocardial perfusion reserve (MPR) is calculated as a ratio between stress MBF/rest MBF. The distinctive perfusion defect seen with CMD is circumferential hypointensity of the endocardium compared to the epicardium (Figure 1); this defect is 4.5 times more readily appreciated when using quantitative perfusion colour maps compared to raw greyscale images [71].

**Figure 1 medicina-59-01596-f001:**
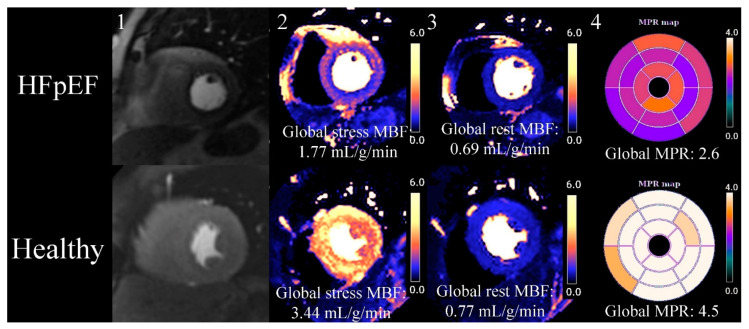
Quantitative myocardial perfusion CMR (cardiovascular magnetic resonance) map of HFpEF (heart failure with preserved ejection fraction) vs. healthy control. This is an exemplar case comparing quantitative myocardial perfusion [78] CMR between an HFpEF case and a healthy control. 1. Stress proton density images at the mid-short-axis level showing the passage of GBCA in the LV. The HFpEF case has a faint circumferential subendocardial hypointensity when compared to the healthy control. 2. Stress MBF is reduced in HFpEF compared to the control, and visual review of HFpEF case shows an overt circumferential subendocardial perfusion defect when compared to the healthy control. 3. Rest MBF, in this case, does not show a significant difference between HFpEF and the healthy control. 4. A bullseye map using a 16-segment model of all 3 short-axis slices (not shown in figure) shows reduced MPR (myocardial perfusion reserve) in HFpEF compared to the healthy control.

Numerous studies (Table 1) consistently demonstrate that those with HFpEF have impaired MPR, making it a potentially important biomarker of the disease. Kato et al. [80] used phase-contrast cine imaging of the coronary sinus to estimate the global MBF and MPR values (synonymous with coronary flow reserve as used in their manuscript) in a cohort consisting of: HFpEF, LVH secondary to hypertension, and healthy controls. MPR was significantly (*p* < 0.05) lower in the HFpEF group (2.21 ± 0.6) when compared to hypertensive (3.05 ± 0.7) and healthy controls (3.83 ± 0.7). At-rest MBF was significantly elevated in HFpEF and hypertension when compared to controls, possibly reflecting the increased resting energy demands in these conditions but also their increased left ventricular mass, as the authors did not adjust for that in their estimation of MBF. Also, stress MBF was lower when compared to controls, further reducing MPR.

In another age- and sex-matched cohort, those with HFpEF were found to have significantly reduced global MPR when compared to healthy controls (2.29 ± 0.6 vs. 3.38 ± 0.8, *p* < 0.01) and higher resting MBF, and lower stress MBF when using a quantitative first-pass perfusion sequence [81]. The cohorts had no significant difference in indexed left ventricular mass, highlighting that importance of assessing the functional component of HFpEF and not relying solely on structural abnormalities.

The hallmark features in HFpEF of impaired MPR may reflect a combination of elevated rest MBF, indicating higher oxygen demand and reduced stress MBF, demonstrating structural and functional defects in the vascular system.

The above studies have examined global values for MBF and MPR; however, the latest iteration of quantitative perfusion sequences [78,79] utilises CMRs superior spatial resolution to automatically segment the heart into not just the regional 16-segment model but further subdivide each segment into an endocardial and epicardial component. This allows the quantification of an endocardial/epicardial ratio that can quantify the degree of inducible subendocardial ischaemia, which has long been suggested as a risk factor for HF and MACE, not just in HFpEF but across cardiovascular diseases [71].

Markley et al. [71] recently reported a 100% prevalence of a stress endocardial/epicardial ratio <1.0 (0.87 (0.81–0.90)) in a carefully phenotyped group (*n* = 19) of obese women with HFpEF and unobstructed coronaries. In relation to diastolic dysfunction, there was a correlation between the stress endocardial/epicardial ratio vs. E/e’ (r = −0.54, *p* = 0.014) and e’ (r = +0.53, *p* = 0.019), indicating that microvascular dysfunction, predominantly of the subendocardium, correlates with diastolic dysfunction, albeit the direction of the relationship is unclear. This finding corroborates with models and invasive study [69,70], showing that a reduced endocardial/epicardial ratio correlates with left ventricular filling pressures and echocardiography cardiographic measures of diastolic function.

Not only are perfusion abnormalities prevalent in HFpEF, they relate to long-term outcome. Visually assessed perfusion defects are independently associated with MACE (HR 5.6 (3.6–8.6), *p* < 0.001) in a large (*n* = 1203) single-centre retrospective cohort [82]. This finding was replicated in a smaller (*n* = 101) prospective cohort [27] utilising quantitative first-pass perfusion techniques, where an MPR < 2.0 was taken as abnormal, and on multivariate analysis was independently predictive of MACE. Larger prospective multicentre studies are required to investigate and confirm this finding.

Despite advances, there are significant challenges with quantitative perfusion CMR that have limited its widespread clinical adoption; it is not included in any major society guidelines and not available in many nonacademic centres. On a practical level, while the output can be “in-line” and available for review before the end of the scan, time investment is required for quality control of the output (Table 2). The person reporting scans should review adequacy of motion correction, cardiac gating, segmentation, and degree of hyperaemia, which can be as insufficient in 10% [83]. Abnormalities in any of the above can cause errors in accurate MBF calculation. Furthermore, there are not yet accepted normative values for MBF, due to a lack of large-scale datasets, differences in models used for estimation of MBF, sequence design, and intervendor differences.

### 2.11. Tissue Characterisation

One of the unique capabilities of CMR is to evaluate diffuse myocardial fibrosis (DMF) using T1 mapping and focal fibrosis with LGE sequences. A study of HFpEF patients [84] using autopsy findings revealed that increasing levels of DMF weakly correlates (r= −0.26, *p* = 0.004) with a reduction in capillary density, indicating that there is an association between MBF and DMF. On CMR, quantification of DMF is achieved through T1 mapping that pre- and post-GBCA can separate the cellular (cardiomyocytes, fibroblasts, endothelial, and red blood cells) from the extracellular (extracellular matrix, blood plasma) compartments, allowing calculation of the extracellular volume fraction (ECV) [85]. Loffler et al. [81] demonstrated higher extracellular volume fraction (ECV) in HFpEF compared to controls (29% vs. 25%, *p =* 0.02), with ECV negatively correlating with MPR (r= −0.54, *p* < 0.01), which mirrors the histological correlation of increasing DMF with decreased capillary density [84]. A similar negative correlation between ECV and MPR was demonstrated by Arnold et al. [27], albeit without statistical significance. This apparent discordance is possibly related to differences in sample selection and imprecise estimates.

Patients with HFpEF are 10 times more likely to have the presence of nonischaemic focal fibrosis (on LGE) when compared to controls [27]; however, the absolute mass of focal fibrosis is not significantly elevated nor was there an association with MPR.

Although tissue characterisation using CMR provides insight into the potential mechanisms and associations with microvascular disease, the findings from initial studies are not as consistent as for myocardial perfusion CMR. Firstly, there is a current lack of evidence base, and large multicentre prospective studies are required to fully assess the associations between DMF and focal fibrosis on diagnosis, monitoring, and prognosis for HFpEF. For these studies to be successful, trial design will be paramount to overcome the limitations of tissue characterisation via CMR. It is accepted that resolution is limited with a single voxel on CMR incorporating millions of myocytes [86]; therefore, the diffuse patchy changes on histology may be difficult to detect, especially in the early stages, due to their diffuse nature that is averaged out across the voxel. There is poor harmonisation of tissue mapping sequences [87], with similar T1 mapping sequences having numerical variation up to 11-fold in numerical value and even applying the same sequence at various sites having potentially significant intersite variability.

Identification and quantification of LGE has significant inter- and intra-observer variability, especially when assessing nonischaemic patterns of LGE [88], which has a significant impact on both single-timepoint and longitudinal studies. Finally, the constant through plane motion of the heart coupled with acquiring sequences over multiple heart beats requires accurate ECG-gating, breath-holding, and motion correction is a persistent challenge and abnormalities in any of the above reducing reliability and introducing artifact.

### 2.12. MRI Assessment of CSVD

Accurately characterising CSVD using MRI presents diagnostic challenges that are like those encountered in HFpEF. Nonetheless, MRI has been in use for a longer time and is more deeply integrated into both clinical investigation and research on CSVD. Neuroimaging features associated with CSVD are diverse and nonspecific and are also seen in nonvascular pathologies such as multiple sclerosis or neuroinfection. Inconsistencies in image acquisition protocols, interpretation, and reporting are being tackled by the Standards for Reporting Vascular Changes on Neuroimaging (STRIVE-1 and STRIVE-2) [32,33], which have been devised and advise on reporting standards to each neuroimaging feature and a list of minimal essential sequences (T1- and T2-weighted sequences, diffusion-weighted imaging, T2-weighted Fluid-Attenuated Inversion Recovery, and T2*-weighted gradient echocardiography) required. These neuroimaging features mostly are unifocal or multifocal morphological tissue abnormalities that represent the consequences to the microvascular disease. Previously accurate and reproducible segmentation of these neuroimaging features, such as WMH and perivascular spaces, was manual and therefore time-consuming. Recently, artificial intelligence approaches [89,90] have brought improved precision and time savings to this process. Similar generalised imaging and reporting standards [91,92] exist for CMR; however, specific recommendations or position papers for HFpEF do not.

Building upon STRIVE, there has been the development of multiple scoring systems for CSVD [93,94], in which the presence of a typical neuroimaging feature scores points that improve prediction of cognitive impairment, recurrent strokes, and all-cause mortality. Similar non-CMR-based scoring algorithms [95] have been validated in HFpEF; however, these are typically complex and require multiple steps, which presents an untapped opportunity to potentially enhance diagnosis and prognosis in HFpEF by incorporating CMR.

Brain MRI has several technical advantages over CMR, which allow for improved assessment of brain structure and tissue. Firstly, brain MRI has superior spatial resolution compared to CMR, as it allows for the application of a narrower field of view to the skull when compared to the thorax. Secondly, though internal pulsations exist inside the brain, brain MRI does not have to contend with cardiac motion, which increases precision and reliability and enables superior tissue characterisation compared to CMR. Additionally, no ECG-gating is required, allowing for faster acquisition or the application of longer sequences that would be prohibitively prolonged in CMR. However, complete coverage of the larger brain can be time-consuming and sensitive to minor patient motion, especially as some of the neuroimaging markers are very small and therefore susceptible to patient motion [96].

The previous paragraphs outline some of the advances and advantages of brain MRI over CMR that are relevant to the assessment of CSVD and HFpEF. A comprehensive discussion of standard sequences outlined by STRIVE and the typical neuroimaging features of CSVD is too broad and beyond the scope of this review and can be obtained elsewhere [28,97]. This section of the review will instead focus on exploring the similarities and differences between CSVD and HFpEF in relation to cerebral and myocardial perfusion and recent advances in cerebral tissue characterisation and its correlations with CMR.

### 2.13. Cerebral Perfusion

Perfusion abnormalities are a common finding in CSVD, with an probable inverse relationship between CBF and CSVD severity [98]. There is ongoing debate as to whether reduced CBF is a cause or consequence [38] of CSVD, as loss of neural tissue could result in decreased metabolic demand and therefore reduced CBF; conversely, reduced perfusion may lead to tissue damage. Especially deeper brain structures, following the trajectories of the inner blood supply, and so-called water shed regions between different vascular territories, such as periventricular white matter, are prone to hypoperfusion. Small-scale human study [99] and animal modelling [100] suggest that perfusion abnormalities can be present prior to tissue changes; however, larger-scale longitudinal studies are required to address this relationship further.

Note that the abovementioned neuroimaging findings are focal morphological abnormalities (“tip of ice-berg” features) of the brain, while the underlying pathophysiological processes are likely more widely and diffusively spread. To asses more global brain regions, dynamic susceptibility contrast (DSC) [101] and arterial spin labelling (ASL) [102] are common validated techniques for estimated cerebral CBF with MRI. DSC measures the first pass of paramagnetic intravascular contrast (such as GBCA), more or less comparable to myocardial perfusion CMR, while ASL utilizes magnetically labelled arterial blood water as an endogenous contrast agent. ASL is advantageous for repeated studies, helping to address (largely theoretical) concerns about repeated administration GBCA and cognitive impairment [103,104]. Note that ASL only sufficiently assesses the grey matter and is prone to the arterial transit time, which varies with macrovascular disease or ageing, while DSC is sufficiently sensitive to measure (the less perfused) white and grey matter.

Meta-analysis [50] combining modalities (MRI, CT, PET) demonstrated that patients with a higher burden of WMH had lower global CBF when compared to patients with lower WMH burden. Lower CBF was observed in both in most grey and white matter areas. The white matter is of crucial importance in CSVD as this region of the brain is exclusively supplied by small vessels of the inner supply system that are susceptible to damage and has the lowest regional perfusion in the brain [98]; therefore, focussing on this region is important as reductions in CBF could be an early sign of CSVD before tissue changes are seen on MRI. Deep WMH are typically associated with vascular degeneration; however, periventricular WMH are a consistent feature in ageing and may be related to mechanical loading of the ependymal cells present at the brain–fluid interface at the ventricular wall [105].

The reduction in resting CBF contrasts with the mildly elevated resting MBF detected in HFpEF, which is possibly explained by fundamental functional differences in the heart and brain in relation to perfusion. The brain undergoes neurovascular coupling, which refers to the coordination between neuronal activity and CBF; when neurons are activated, they demand increased oxygen and nutrients, which triggers an increase in CBF to that specific region of the brain. In CSVD, once tissue damage has occurred and neurovascular coupling is impaired [106], CBF may not properly follow the metabolic demand of the brain tissue.

Although HFpEF and CSVD may differ in response to basally required blood flow, they share a significant similarity when it comes to measuring endothelial function in the form of cerebral vascular reserve (CVR) [51]—the ratio between stress and rest CBF analogous to MPR in the heart. As endothelial dysfunction is likely a key pathophysiological mechanism in the development of CSVD, assessing it through CVR measurement may be advantageous in CSVD evaluation. “Stressing” the brain is usually achieved through inducing hypercapnia (via breath-holding or breathing CO_2_-enriched air) [51,98] or administration of acetazolamide; both cause temporary vasodilatation of the cerebral microvascular system and an increase in CBF.

As seen in HFpEF, CVR in CVSD is often reduced when compared to controls. Thrippleton et al. [107] demonstrated lower grey-matter CVR in patients with minor stroke when compared to controls using a hypercapnic CO2-enriched air protocol. There is likely an inverse association between CVR and CSVD severity on neuroimaging, with a higher WMH volume being associated with lower white-matter CVR [108]. CVR measurement is also promising to untangle the cause or consequence debate for the development of WMH as impaired CVR can be detected in areas of normal-appearing white matter that progress to WMH [109].

Although impaired CVR is a promising biomarker in detecting early CSVD before tissue changes and correlates with traditional markers of CSVD severity—presenting a promising parallel with impaired MPR seen in HFpEF—the studies mentioned above have similar limitations as discussed with myocardial perfusion CMR. Many of them are small-scale and single-site, with limited longitudinal follow-up, and there is considerable diversity in the mechanism of inducing stress, image acquisition, and postprocessing. Therefore, although the evidence base is promising, it remains thin, and significant expansion is necessary before drawing firm conclusions.

### 2.14. Vessel Size Imaging

Vessel size (and density) imaging is a research technique that can provide in vivo quantitative estimates of mean vessel diameter and density within a given voxel and is sensitive to large vessels and microvasculature <10 µm in diameter [110]. It exploits the difference in the ratio of relaxation rate changes as measured by gradient and spin echocardiography pulse sequences on T2- and T2*-weighted images during the passage of intravascular contrast, and if hybrid sequences combining both spins are employed, it can also estimate CBF, thereby making it an attractive combination sequence. Thus far, most of the research for vessel size imaging has been in the field of cerebral neoplasms [110]; however, initial study [111] in patients with CSVD detected a significantly increased mean vessel diameter when compared to matched controls. The finding of increased mean vessel diameter indicates that there is a potential reduction in the microvessels that increases the overall mean vessel diameter in the brain as the large vessels are preserved. This suggests that not only is there microvascular dysfunction (as evidenced by impaired CVR) but also reduced microvascular density that has also been noted in HFpEF autopsy study [84].

As with other techniques vessel size imaging is in its infancy for CSVD and further research is required; also, whether this technique can be translated into the heart and overcome motion is to be seen but could be of value.

### 2.15. Blood–Brain Barrier Integrity

The BBB refers to the properties of the microvasculature of the brain preventing easy exchange between the blood and the tissues, of which there is no analogue inside the myocardium. Pericyte endothelial cells tightly regulate the movement of molecules, ions, and cells between the blood and the brain [38]. As the integrity of the capillaries is disrupted in CSVD, this may cause BBB leakage that can be detected using dynamic contrast-enhanced (DCE) MRI. DCE-MRI uses T1-weighted images that are acquired pre- and ~15 min post-GBCA administration, and the T1 longitudinal relaxation time (that is shortened by GBCA) is measured allowing quantification of BBB leakage from a pharmacokinetic model [38]. Increased BBB leakage is associated with CSVD, reduced perfusion, and cognitive impairment [112,113]. This finding is akin to the increased ECV observed in HFpEF when employing the same principle of T1 longitudinal shortening pre- and post-GBCA administration to gauge the quantity of GBCA in the extracellular space, where increased levels are deemed pathological.

The main limitations of the technique are the requirement for GBCA, prolonged delay required between GBCA administration and acquisition (>15 min), and the low signal-to-noise ratio of the leakage effect [38]. Newer techniques [114,115] have been developed targeting the water exchange across the endothelium that do not require contrast material.

### 2.16. Diffusion Tensor Imaging

Diffusion tensor imaging (DTI) is an advanced imaging technique used to assess the microstructural organization of a tissue by examining the diffusion of water molecules [116]. It provides information about the degree of alignment (fractional anisotropy) and overall diffusion (mean diffusivity) within the targeted tissue. DTI is chiefly directed at the white matter tracts as these are believed to play a crucial role in cognitive function. Patients with CSVD, when compared to controls [116,117], have lower fractional anisotropy and increased mean diffusivity, indicating loss of alignment of the white matter tracts as water diffusion is less restricted, and these DTI measures correlate with cognitive impairment [116]. While showing promise in cardiac applications [86], the implementation of this technique in research is relatively recent.

## 3. Discussion

The comparison between HFpEF and CSVD is becoming increasingly relevant, with some experts suggesting that HFpEF could be considered a “dementia of the heart”, [118] with both being viewed as manifestations of maladaptive cardiac or brain ageing. Both HFpEF and CSVD pose significant ongoing challenges due, in part, to being classified as a single large heterogenous category with a shared similar aetiology. However, they consist of multiple distinct phenotypes, making it difficult to differentiate and target them effectively. Greater attention is required to discern and focus on specific subtypes within these conditions. Fundamentally, microvascular disease is characterized by a disturbance in the structure and function of the microvessels, resulting in a mismatch between supply and demand and subsequent damage to the end organs.

Insights from animal models and autopsy studies [14,16,18,84,119] indicate that both HFpEF and CSVD have some shared pathophysiology. In the case of HFpEF, it is hypothesized that endothelial dysfunction leads to structural and functional abnormalities in the coronary microvasculature including: small artery remodeling and impaired vasodilation, capillary rarefaction and other capillary structural abnormalities. This is associated with abnormal cardiomyocyte function, energetics, and diffuse myocardial fibrosis. This is mirrored in CSVD, where microvascular disease of the cerebral arteries is characterized by endothelial dysfunction, small artery remodeling, blood–brain barrier disruption, capillary structural abnormalities, and capillary rarefaction. There is associated cerebral parenchymal damage with demyelination, neuronal loss, oligodendrocyte damage, and axonal injury.

In both, endothelial dysfunction leading to the disruption in nitric oxide pathways has been identified as a potential common pathway that may be a suitable therapeutic target. Recent CSVD phase 2 study [120] indicates that Isosorbide Mononitrate and Cilostazol, both of which modulate endothelial function, may improve long-term outcome in lacunar stroke. Moreover, they share similar risk factors including aging and pro-inflammatory cardiovascular risk factors such as diabetes mellitus, hypertension, and obesity. Despite advances in research, there is still a shortage of appropriate animal models of disease which may be reflective of HFpEFs and CSVDs heterogenous nature [51].

Both HFpEF and CSVD face research gaps due to the inability to spatially resolve the microvasculature in vivo. This is critical in the early stages when organ tissue changes and clinical symptoms are not yet present. Functional imaging techniques are emerging as a promising approach to overcome these challenges and provide insights into microvascular functional changes before tissue changes occur.

Quantitative perfusion CMR shows promise in reliably assessing the distinctive circumferential subendocardial pattern of impaired MPR in HFpEF [71]. Incorporating this technique into diagnostic criteria and therapeutic trials could refine understanding of HFpEF and potentially reduce time and costs associated with large clinical trials.

Impaired CVR may indicate early CSVD before irreversible tissue damage [109]. This finding adds to the growing body of evidence supporting endothelial dysfunction in HFpEF and CSVD. Stress protocols for assessing CVR are less established with added complexity of neurovascular coupling and the unique structural composition [98]. To advance in this field, widely available validated protocols are required, along with large-scale multicenter trials that are planned [121].

Comparatively, quantitative perfusion CMR is ahead due to the relative ease of stressing the heart and stress cardiac assessment being a fundamental test of cardiac function with extensive evidence base. It is limited by several factors including: limited availability, lack of standardization, and no formally accepted normative values [77].

Brain MRI excels in tissue characterization, and while this review has focused predominantly on the perfusion changes associated with HFpEF and CSVD, the tissue response is an essential component and cannot be neglected. Future CMR research requires continued effort to characterize myocardial tissue in HFpEF. Given that HFpEF and CSVD may be results of systemic microvascular disease, future research could aim to assess other organs susceptible to microvascular disease or target capillary beds that can be visualized in vivo, such as the retina or sublingually [122]. Microvascular retinal abnormalities have been identified in both cardiovascular [123] and neurological [124,125] conditions and may provide an additional avenue to noninvasively assess for microvascular disease.

## 4. Conclusions

Both HFpEF and CSVD pose significant challenges in terms of understanding pathophysiology, accurate diagnosis, and development of effective therapies. MRI offers a promising approach for the comprehensive assessment of these disorders, due to its unique noninvasive tissue characterization properties and emerging role in heart and brain functional perfusion imaging. Despite MRI’s potential, there are still limitations that need refinement before translation to clinical practice. MRI needs to be part of a multiparametric clinical assessment, and future research needs to also consider the tissue response to microvascular disease to obtain a complete overview.

## Figures and Tables

**Table 1 medicina-59-01596-t001:** Imaging features of CSVD and pathological mechanisms: [30,31].

Imaging Feature	Description and Pathophysiology	Clinical Image	MRI Findings
Recent small subcortical infarct (formerly lacunar infarct):	Occlusion of perforating artery causing distal infarction of brain parenchyma [28]. Makes up ~25% of acute ischaemic strokes.	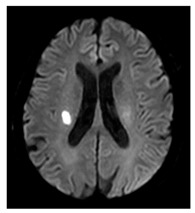	≤20 mm Best identified on DWI Hyperintense: T2, FLAIR, DWI Hypointense: T1 Isointense: T2*-GRE
White matter hyperintensity:	White matter demyelination resulting from multiple pathological insults including: chronic hypoperfusion, blood–brain barrier leakage, impaired amyloid clearance, and iron deposition [28]	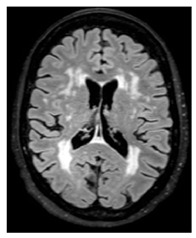	Variable in size Hyperintense: FLAIR, T2, and T2*-GRE Hypointense: T1 Isointense: T1
Lacune:	CSF-filled cavity within the basal ganglia or white matter that is presumed to arise from prior small deep brain infarction [32]	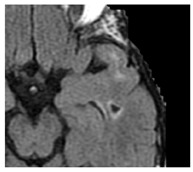	3–15 mm Best distinguished on FLAIR with hypointense centre and hyperintense rim Hyperintense: T2 Hypointense: T1, DWI Isointense: DWI, T2*-GRE
Enlarged perivascular spaces:	Fluid-filled compartments surrounding the small blood vessels in the brain that allow clearance of waste metabolites from the brain. Enlargement possibly arises from blockage of the perivascular space leading to accumulation of waste products [33].	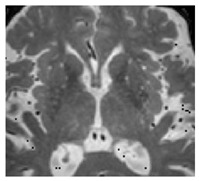	<2 mm Hyperintense: T2 Hypointense: T1, FLAIR Isointense: DWI, T2*-GRE
Cerebral microbleeds:	Perivascular haemosiderin deposits that leak from capillaries, implying the breakdown of the blood–brain barrier and endothelial dysfunction [28]	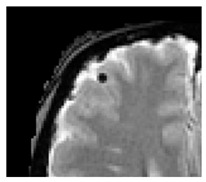	<10 mm Hypointense: T2*-GRE, SWI Isointense: DWI, FLAIR, T2, T1
Cortical cerebral microinfarct:	New additions with STRIVE-2 that are typically visible on microscopic neuropathological examination or high-field (7T) MRI. Some larger cortical cerebral microinfarcts (0.5–4 mm) can be seen on conventional MRI strength.	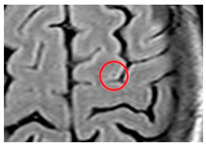	<4 mm Hyperintense: T2, DWI (if acute) Hypointense: T1

CSF—cerebrospinal fluid, DWI—diffusion-weighted imaging, FLAIR—fluid-attenuated inversion recovery, GRE—gradient echo, SWI—susceptibility-weighted imaging.

**Table 2 medicina-59-01596-t002:** CMR studies in HFpEF.

Study	Population	Modality	Perfusion Parameters		Tissue Characteristics		Outcome
Kato et al. 2015 [80]	HFpEF (n = 25) Controls (n = 19) Hypertensives (n = 13)	Phase-contrast CMR	Stress MBF	↓	No tissue characterisation		Not linked to outcomes
			Rest MBF	↑			
			MPR	↓			
Löffler et al. 2019 [81]	HFpEF (n = 19) Controls (n = 15)	Quantitative perfusion CMR	Stress MBF	↓	ECV	↑	Not linked to outcomes
			Rest MBF	↑	LGE presence	Not compared to controls	
			MPR	↓			
Arnold et al. 2022 [27]	HFpEF (n = 101) Controls (n = 42)	Quantitative perfusion CMR	Stress MBF	↓	ECV:	↑	Reduced MPR independently associated with MACE. Increased ECV associated with MACE.
			Rest MBF	↔	LGE presence:	↑	
			MPR	↓			
Pezel et al. 2021 [82]	HFpEF (n = 1203) No control group	Semi-quantitative visually assessed perfusion for segments of ischaemia on CMR	N/A (no control group)		N/A (no control group)		Moderate (3–5 segments) and severe (≥6) segments of ischaemia associated with MACE. Presence of LGE associated with MACE on multivariate regression.

HFpEF: heart failure with preserved ejection fraction, CMR: cardiovascular magnetic resonance, MPR: myocardial perfusion reserve, MACE: major adverse cardiovascular events, ECV: extracellular volume fraction.

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
