# Peer review of "Assessment of Microvascular Disease in Heart and Brain by MRI: Application in Heart Failure with Preserved Ejection Fraction and Cerebral Small Vessel Disease"

_medicina, 2023, doi:10.3390/medicina59091596_

Round 1
Reviewer 1 Report
1. Cortical cerebral microinfarct: please consider a better imaging example, where the infarct clearly involves the cortex.
2. Consider discussing
2.1. Brain Diffusion Tensor Imaging (DTI)
2.2. Automated detection of white matter T2 hyperintensities
2.3. Automate calculation of perivascular spaces: Lynch KM, Sepehrband F, Toga AW, Choupan J. Brain perivascular space imaging across the human lifespan. Neuroimage. 2023 May 1;271:120009. doi: 10.1016/j.neuroimage.2023.120009. Epub 2023 Mar 11. PMID: 36907282; PMCID: PMC10185227.
2.4. CSVD grading systems such as Manolio, Fazekas and Schmidt, Scheltens, “total SVD score”, and RCSVD. Hazany S, Nguyen KL, Lee M, Zhang A, Mokhtar P, Crossley A, Luthra S, Butani P, Dergalust S, Ellingson B, Hinman JD. Regional Cerebral Small Vessel Disease (rCSVD) Score: A clinical MRI grading system validated in a stroke cohort. J Clin Neurosci. 2022 Nov;105:131-136. doi: 10.1016/j.jocn.2022.09.014. Epub 2022 Sep 29. PMID: 36183571; PMCID: PMC10163829.
3. Consider discussing histopathological studies of brain and heart with small vessel disease.
4. Consider citing current studies on retina and small vessel disease at the end of discussion.
Author Response
- Cortical cerebral microinfarct: please consider a better imaging example, where the infarct clearly involves the cortex.
Thank you for this comment, we have reached out within our team to try and find a more suitable image however due to the nature and size of cortical cerebral microinfarcts we haven’t been able to find a better example.
- Consider discussing
2.1. Brain Diffusion Tensor Imaging (DTI)
Thank you for your comment, this was a topic that was discussed prior to submission and initially not included due to concerns of length of the article. We agree that DTI imaging is of value of discussion and additionally it has recently been implemented in the research sphere of cardiology making its discussion more pertinent. We have included a short section below discussing its merits.
Diffusion tensor imaging:
Diffusion tensor imaging (DTI) is an advanced imaging technique used to assess the microstructural organization of a tissue by examining the diffusion of water molecules.118 It provides information about the degree of alignment (fractional anisotropy) and overall diffusion (mean diffusivity) within the targeted tissue. DTI is chiefly directed at the white matter tracts as these are believed to play a crucial role in cognitive function. Patients with CSVD, when compared to controls,118,119 have lower fractional anisotropy and increased mean diffusivity indicating loss of alignment of the white matter tracts as water diffusion is less restricted and these DTI measures correlate with cognitive impairment.118 While showing promise in cardiac applications,88 the implementation of this technique in research is relatively recent and not been rigorously applied to HFpEF.
2.2. Automated detection of white matter T2 hyperintensities
Thank you for the comments on this, appropriate section has been entered for both white matter hyperintensities and perivascular space segmentation using automated methods.
“Previously accurate and reproducible segmentation of these neuroimaging features, such as WMH and perivascular spaces, was manual and therefore time consuming. Recently artificial intelligence approaches91,92 have bought improved precision and timesaving to this process.”
2.3. Automate calculation of perivascular spaces: Lynch KM, Sepehrband F, Toga AW, Choupan J. Brain perivascular space imaging across the human lifespan. Neuroimage. 2023 May 1;271:120009. doi: 10.1016/j.neuroimage.2023.120009. Epub 2023 Mar 11. PMID: 36907282; PMCID: PMC10185227.
See above section regarding automated detection of white matter T2 hyperintensities
2.4. CSVD grading systems such as Manolio, Fazekas and Schmidt, Scheltens, “total SVD score”, and RCSVD. Hazany S, Nguyen KL, Lee M, Zhang A, Mokhtar P, Crossley A, Luthra S, Butani P, Dergalust S, Ellingson B, Hinman JD. Regional Cerebral Small Vessel Disease (rCSVD) Score: A clinical MRI grading system validated in a stroke cohort. J Clin Neurosci. 2022 Nov;105:131-136. doi: 10.1016/j.jocn.2022.09.014. Epub 2022 Sep 29. PMID: 36183571; PMCID: PMC10163829.
Thank you for highlighting these references, I have expanded the section on scoring systems to highlight recent work by Hazany et al.
“Building upon STRIVE there has been the development of multiple scoring systems for CSVD,95,96 in which the presence of a typical neuroimaging feature scores points that improves prediction of cognitive impairment, recurrent strokes, and all-cause mortality.”
- Consider discussing histopathological studies of brain and heart with small vessel disease.
We appreciate the attention for the limited coverage of histopathological studies in this review. While we acknowledge the potential significance of including such discussions, given the current length of the review and extensive nature required for in-depth histopathology analysis, we believe that it falls outside the scope of this specific review.
- Consider citing current studies on retina and small vessel disease at the end of discussion.
Please find the addition of some recent relevant studies in retinal imaging and cardiovascular and neurological disease cited in the discussion.
“Microvascular retinal abnormalities have been identified in both cardiovascular124 and neurological125,126 conditions and may provide an additional avenue to non-invasively assess for microvascular disease.”
Blair GW, Stringer MS, Thrippleton MJ, et al. Imaging neurovascular, endothelial and structural integrity in preparation to treat small vessel diseases. The INVESTIGATE-SVDs study protocol. Part of the SVDs@Target project. Cereb Circ Cogn Behav. 2021;2:100020. doi:10.1016/j.cccb.2021.100020
van Dinther M, Bennett J, Thornton GD, et al. Evaluation of miCRovascular rarefaction in vascUlar Cognitive Impairment and heArt faiLure (CRUCIAL): Study protocol for an observational study. Cerebrovasc Dis Extra. 2023;13(1):18-32. doi:10.1159/000529067
126. Diaz-Pinto A, Ravikumar N, Attar R, et al. Predicting myocardial infarction through retinal scans and minimal personal information. doi:10.1038/s42256-021-00427-7
Reviewer 2 Report
Dears,
The paper has well-designed research methods, appropriate statistical analysis and a relatively good interpretation of the results.
-Please be sure to use only keywords accordingly to medical subject headings (Mesh word) for a better indexing.
I suggest you add a table with the list of abbreviations used in the text.
I suggest you implement the abstract in order to make it more understandable to authors.
The introduction should be expanded perhaps by adding a section on temporomandibular disorders. I recommend some references:[10.3390/jcm12072652];[10.1111/joor.13496]
The conclusion is in accordance with the objectives of the research, its results and their interpretation, as well as the relevant literature.
Regards
Dear,
Punctuation and grammatical errors are present within the text. Please correct them
Regards
Author Response
The paper has well-designed research methods, appropriate statistical analysis and a relatively good interpretation of the results.
-Please be sure to use only keywords accordingly to medical subject headings (Mesh word) for a better indexing.
I suggest you add a table with the list of abbreviations used in the text.
I suggest you implement the abstract in order to make it more understandable to authors.
The introduction should be expanded perhaps by adding a section on temporomandibular disorders. I recommend some references:[10.3390/jcm12072652];[10.1111/joor.13496]
The conclusion is in accordance with the objectives of the research, its results and their interpretation, as well as the relevant literature.
Thank you for your review on comments on the review article. I have reviewed the keywords and they related to established MeSH words to allow for appropriate indexing of the article. A table containing abbreviations was available at the end of the review article, for clarity I have moved it to beginning of the article for clarity to the reader. An abstract was implemented on page 1 of the review article. Thank you for the input and suggested references for the addition of temporomandibular disorders. We have discussed this within our group and disagree that this is pertinent to subject matter and have elected to not include this within our submitted manuscript.
Round 2
Reviewer 1 Report
No comments